# How Do Tougher Plastics Ban Policies Modify People’s Usage of Plastic Bags? A Case Study in China

**DOI:** 10.3390/ijerph182010718

**Published:** 2021-10-13

**Authors:** Bairong Wang, Yuhua Zhao, Yong Li

**Affiliations:** 1School of Economics and Management, Shanghai Maritime University, Shanghai 201306, China; bairongw@buffalo.edu (B.W.); 15655326275@163.com (Y.Z.); 2School of Marxism, Shanghai Maritime University, Shanghai 201306, China

**Keywords:** plastic waste crisis, plastic bags, China’s plastics ban policies, policy execution loopholes, spill-over effects

## Abstract

This study analyzes people’s bag usage before and after the introduction of the tougher 2021 plastics ban policies by counting the number of charged carrier bags, inner bags, old plastic bags, and reusable bags used by consumers at the exits of the investigated supermarkets in Shanghai, China. The results indicate positive effects of the tougher 2021 measures by observing significantly decreased usage of charged carrier bags by 46%, and significantly increased usage of old plastic bags and reusable bags by 117% and 36%, respectively. Policy execution loopholes are found in some supermarkets which do not follow the tougher plastics ban measures. Fortunately, the spill-over effects from tougher-measure-executing supermarkets fix this issue to some extent. Additionally, the tougher 2021 measures fail to be the most powerful impacting factor on people’s usage of each type of bag. To produce better plastics reducing results, other bag-targeted measures are necessary.

## 1. Introduction

Nowadays, people cannot be more familiar with plastic bags due to our life’s heavy dependence on them, which are lightweight, water-resistant, and convenient to use [1,2]. However, plastic bags have also been blamed as one of the worst inventions in human history given the devastating damage they have caused [3,4]. Hundreds of species are affected by plastic bags and each year, millions of sea birds, seals, turtles, and other maritime creatures mistake plastic bags for food and unfortunately die of this mistake [5]. Ill-managed plastic bags are caught in trees and therefore destroy the natural beauty of landscapes and increase the cleaning cost [6]. Within under-developed countries or areas (e.g., slums), the problem is more serious due to the poor solid waste management there [7]. Worse still, to naturally degrade, plastic bags need 500–1000 years, but their use time is simply 12 min on average [8]. The alarming damage caused by plastic bags has induced increasingly world-wide efforts to curb their intensive usage [2,9]. Levies, fees, and legislations are among the most popular instruments that have been used so far. The well-known levy policy from Ireland has been proven to be effective in reducing people’s usage of plastic bags by 94% when introduced in 2002 [10]. Portugal also introduced a similar tax in 2015 to reduce people’s plastic carrier usage [11]. Additionally, studies are also conducted to evaluate the effectiveness of plastic ban policies [12,13], to analyze people’s attitudes towards and using motivations and barriers of plastic bags [14,15,16] and to investigate and predict people’s usage of plastic bags [2,17,18,19].

China, with the largest population in the world, has been active in cracking down on the plastics crisis. After the first introduction of national plastics ban law in 2008, the Chinese government introduced new plastics ban policies in January 2020 and required all supermarkets in major cities stop using non-biodegradable plastic carrier bags by the end of 2020 [20,21]. One year later, other tougher plastics ban measures were introduced in Shanghai, China, which completely forbade the usage of plastic carrier bags and required all supermarkets to sell only cloth or nylon carrier bags priced from RMB 1.0 to 39.0. Retailers violating the rules will be fined RMB 10,000–100,000 according to the Law on the Prevention and Control of Environmental Pollution by Solid Waste, which was revised in April and came into effect in September 2020 [22,23]. As of this study, measures for plastics management in China are all penalty-oriented. According to Gray’s theory of personality, there exists a behavioral activation system (BAS) that controls people’s sensitivity to penalty and avoidance motivation [24]. Although the penalties of new tougher laws did not target consumers, as price serves as one of the most important determinants in consumers’ purchase motivation [25], the forbidden high price for charged carrier bags may work similarly to a penalty. Viewing this, we are interested to know how consumers respond to the new 2021 plastics ban policies, and whether the policies work effectively in modifying people’s bag-using behaviors. Therefore, we conduct a field study by counting people’s usage of different types of bags within supermarkets. Results of this study could provide valuable insights on policy effectiveness and offer potential constructive suggestions on plastics crisis management.

## 2. Method

### 2.1. Data Collection

This study makes a field study of people’s bag usage within supermarkets in Shanghai, China before and after the introduction of tougher 2021 plastics ban measures. Specifically, 9 Shanghai-located supermarkets from different popular chain brands were visited for data collection on weekdays from July to August 2020 and late January to February 2021. The visiting time is from 17:30 to 19:00, when middle-young consumers are off from work and available for shopping. For each supermarket, we repeated our data collection three times to reduce the potential bias caused by counting errors or randomness. A recorder would stand at the exit of each supermarket and count the number of 4 different types of bags, including charged carrier bags, plastic inner packaging bags (inner bags hereafter), old plastic bags, and reusable bags. Before our official data collections, we conduct a pilot test in one supermarket to train the two recorders regarding the counting rules. The Cohen’s α value for counting reliability comes to a satisfied value of 0.82. Considering the paramount impact of demographic factors on people’s plastic bag usage [18], we record their genders and briefly categorize the consumers into two generations, old and middle–young by inferring whether their age is over 60 or not. Among the 9 investigated supermarkets, three are grocery supermarkets located in Shanghai’s Yangpu and Pudong districts; six are non-grocery ones located in Shanghai’s Yangpu, Hongkou, and Pudong districts. Grocery supermarkets are those selling only limited fresh food products (e.g., fruits and fresh meat) and most of their products are packaged at the point of sale. These markets, such as China’s popular grocery supermarket Blt (Better life together), mainly serve consumers who would like the purchase quick and time-saving, and therefore are more popular among young consumers. On the other hand, the non-grocery supermarkets, such as Walmart, provide a much broader range of products and have most of their fresh products unpacked for selling. In our study, we also categorize these two types of supermarkets to learn how packaging style impacts people’s bag usage. During our investigation, two non-grocery supermarkets did not follow the new tougher 2021 measures at all.

### 2.2. Statistical Analysis

To reduce data collection bias, we first used the Kruskal–Wallis test [26] to examine the difference in data collected in each visit to the same market. Descriptive statistical analysis was used to summarize bag usage patterns in 2020 and 2021, respectively. We then used the KW test again to check whether the tougher 2021 measures could further modify people’s usage of the 4 above-mentioned different types of bags. Finally, logistic regression was used to analyze the influence power of different factors on people’s bag usage, including gender, generation, packaging style, and tougher measures.

## 3. Results

### 3.1. Plastic Bag Usage before and after Tougher Measures

As shown in Table 1, all the KW tests support significant (*p* < 0.05) changes in people’s bag usage. Facing tougher measures, the consumers reduce their purchasing of charged carrier bags by almost 46%. Two options have been made to meet people’s carrying demand. First, they choose to bring significantly more reusable bags (χ^2^ = 124.500; *p* = 0.000) and old plastic bags (χ^2^ = 124.300; *p* = 0.000). The usage of reusable bags has seen an increase of 36%, while that value for old plastic bags surprisingly reaches 117%. Second, they choose to use significantly χ^2^ = 40.705; *p* = 0.000 more free inner bags, but the extent of the increase is just 0.4%.

As mentioned above, two of our investigated supermarkets do not follow the tougher measures at all. As these two supermarkets are non-grocery markets, we compare their dynamic bag usage to that of their four non-grocery counterparts where tougher measures are strictly executed. As shown in Figure 1, within markets either with or without tougher measures, the usage of reusable bags and old bags increases, and the usage of charged bags decreases. Except for the usage of old plastic bags, the changing amount of other bags is smaller within these two markets compared to that within their tougher-measure-executing counterparts. The spill-over effects are found from tougher-measure-executing supermarkets.

### 3.2. Plastic Bag Usage with Different Packaging Styles before and after Tougher Measures

On average, both grocery and non-grocery markets show reduced usage of charged carrier bags by over 40%. Regarding the usage of inner bags, the two types of markets vary in changing directions. As shown in Table 2, the usage of inner bags drops by over 50% within grocery markets but increases by 2.7% within non-grocery markets. As inner bags are rarely available in grocery markets, packaging style may primarily account for the different changes of inner bag usage. Thus, source control is inferred to be effective in curbing the usage of inner bags.

### 3.3. Impacting Patterns of Different Factors on the Usage of Different Bags

We use the techniques of logistic regression to learn how each influential factor modifies people’s usage of different types of bags. We classify people’s bag usage into a binary situation, i.e., to use or not to use. We then use this binary choice as the dependent variable, and the factors of gender, generation, packaging style, and tougher measures as independent variables. We first examine the potential influence of each factor by the KW test and only those significantly influential on people’s bag usage are further analyzed in logistic regression models. As shown in Table 3, the VIF value of each factor is around 1.0 in the regressions models, indicating a multi-collinearity-free status of the independent variables [27]. For charged carrier bags, generation is the most powerful impacting factor, then followed by tougher measures. Regarding old plastic bags, its usage is dominated by tougher measures. As for reusable bags and inner bags, the most powerful impacting factors are gender and packaging style, respectively.

## 4. Discussion

### 4.1. Effectiveness of Tougher Plastics Ban Measures

Generally, the tougher 2021 measures work effectively in modifying people’s usage of different bags by significantly reducing people’s purchase of charged carrier bags, and by significantly increasing people’s usage of reusable bags and old plastic bags. Unavoidably, random data collection errors or other factors may account for the change. However, we still prefer to ascribe this valuable change to execution of the tougher 2021 measures. The reasons are two-fold. Firstly, we collected our data at the same markets and at the same time of a day during our two field studies. The only difference lies in the execution of tougher plastic ban policies. Secondly, our regression results also support the significant impact of the execution of tougher policies on the usage of all types of bags. With the tougher 2021 policies, on the one hand, plastic charged carrier bags are banned, which compels consumers to change; while, on the other hand, only expensive cloth or nylon charged carrier bags are available, which could again force those price-sensitive consumers to seek other cheaper options. Consequently, the consumers reduce their usage of charged carrier bags by nearly 46%, which is similar to the 49% decrease in carrier bag usage observed previously [28], while, as for inner plastic bags, their usage increases significantly as some consumers would take advantage of policy loopholes and turn to free inner plastic bags as alternatives for carrier bags. A similar loophole exploiting behavior is also observed in Portugal where the consumption of plastic garbage bags increases dramatically thanks to the introduction of plastic ban policies [11]. That is to say, as long as loopholes exist, people will exploit them to their best interests. However, what surprises us is the tiny increase in inner bag usage, i.e., 0.4%, which is quite different from observations from existing studies [11,29]. This is partially due to the complete ban of free inner bags within some grocery supermarkets. However, even within non-grocery supermarkets, the increased amount is also just 2.7%. That is to say, the majority decreased usage of charged carrier bags is offset by the increased usage of old bags and reusable bags rather than that of inner bags, which is positive for plastics management. For these positive results, by no means are they achieved simply through the tougher 2021 policies. As shown in our previous study [29], the 2020 policies could boost people’s usage of reusable bags by forbidding the usage of non-biodegradable plastic carrier bags and imposing high prices for biodegradable plastic carrier bags. The tougher 2021 policies continue to strengthen people’s behavioral change that was already shaped by 2020 policies. For example, no-bag shopping and bag-carrying behaviors start to prevail after the introduction of national regulations and the behaviors become more popular and widely accepted in 2021. Therefore, the strengthening effect could in turn explain the tiny increase in inner plastic bag usage in 2021.

Regarding this study, measures for plastics policies in China are all penalty-oriented. Incentives are not available for businesses or consumers to comply with the plastic bag ban policies. To improve the effectiveness of plastics management, future studies are encouraged regarding how various incentives could shape people’s usage of plastic bags.

### 4.2. Spill-Over Effects on Bag Usage from Tougher-Measure-Executing Supermarkets

Spill-over effects are found within the two no-tougher-measure supermarkets by observing similar changes in the usage of different bags. It is possible that the similar changes arise from data collection errors; however, we believe the spill-over effects could explain better. The reasons are three-fold. To begin with, within the two rounds of data collection we keep bag counting rules the same and repeat our data collection for each site three times to keep random collection errors as few as possible. Secondly, except for old plastic bags, the amount of usage change of other bags is much smaller within no-tougher-measure markets compared to that within tougher-measure-executing supermarkets. Thirdly, people are generally consistent in their habits and behaviors [30]. When consumers are forced to bring reusable bags or old plastic bags for shopping in tougher-measure-executing supermarkets, bag-carrying habits could be formed and reinforced.

### 4.3. Loopholes Exist, but Damage Less When Tougher Measures Are Imposed

In this study, loopholes exist not only in policy design, but also in execution. For the policy design loophole, the government does not have their policies targeted at inner bags. Execution loopholes are found as the investigated supermarkets still use plastic carrier bags for their online delivery service, and two of our investigated supermarkets fail to follow the new tougher 2021 rules. Fortunately, side effects of the offline execution loophole could be offset by the spill-over effects from the tougher-measure-executing supermarkets. However, if the behavior change is associated with the external tougher policies rather than intrinsically and environmentally motivated [31], it is fragile and susceptible to disappearance once policies are not well executed. In this regard, loopholes still warrant fixing and making the behavior change intrinsically and environmentally motivated should be the ultimate goal.

## 5. Conclusions

This research is a field study of people’s bag usage in Shanghai, China before and after China’s tougher 2021 plastics ban measures. Our results reveal several valuable insights as follows.

The first key finding is the positive results produced by the tougher 2021 measures. The usage of charged carrier bags has decreased by almost 46%. As the cost for charged carrier bags becomes much higher than before, more people start to use old plastic bags and reusable bags as alternatives. Second, despite the existence of execution loopholes, the spill-over effects from tougher-measure-executing supermarkets could do some fix. As mentioned above, two of our investigated supermarkets do not follow the tougher rules and continue to offer plastic carrier bags. However, surprisingly similar increased usage of reusable bags and old plastic bags, and decreased usage of charged bags are also observed in these two markets, which we ascribe to the spill-over effects from tougher-measure-executing supermarkets.

Finally, our study shows that tougher measures are not always the largest determining factor on people’s usage of each type of bag. For instance, the usage of inner bags is most impacted by packaging style. It is therefore suggested that the demand for free inner bags is elastic as people would naturally reduce their usage when products are packaged for selling. Therefore, simply imposing higher fees and legislation measures would not necessarily produce the best anti-plastics results. In this regard, other bag-targeted measures are necessary.

### Limitation and Future Research

The study is limited by the small sample of supermarkets within Shanghai, China. Consequently, the results of it cannot be representative of the whole situation in China. Further research is encouraged to tackle this limit. Moreover, results of this study are based on objective data of people’s bag usage, therefore what intrinsic motivations or mentalities are behind these usage patterns deserves more future research efforts. Psychology theories could be used to dig deeper into this topic.

## Figures and Tables

**Figure 1 ijerph-18-10718-f001:**
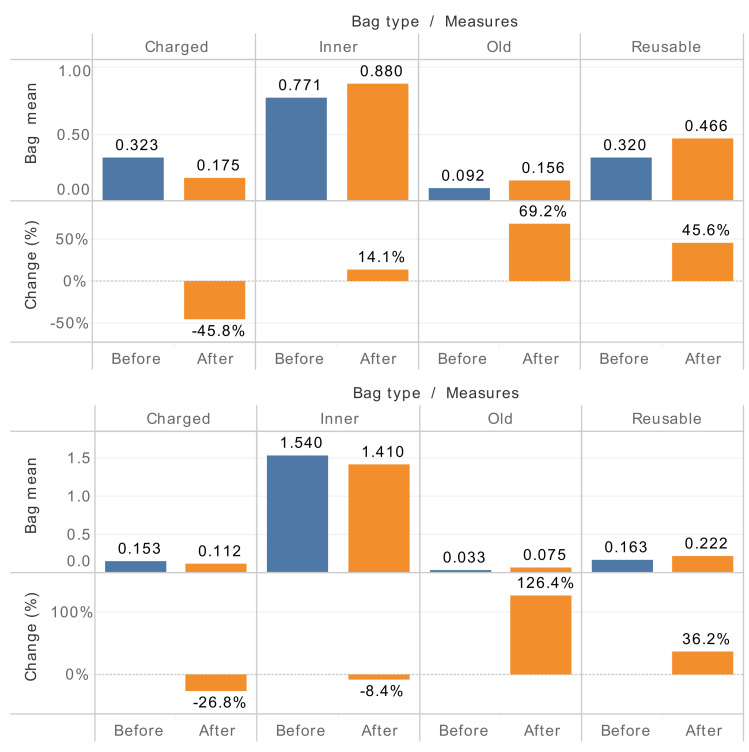
Comparative bag usage of non-grocery markets with (upper) and without (lower) tougher measures.

**Table 1 ijerph-18-10718-t001:** The mean values and KW test results for the usage of different bags before and after tougher measures.

Measure Status	Inner	Reusable	Old	Charged
Before	0.630	0.327	0.094	0.304
After Change	0.633 0.40%	0.446 36.39%	0.204 117.02%	0.165 −45.72%
KW test	χ^2^ = 40.705	χ^2^ = 124.500	χ^2^ = 155.300	χ^2^ = 151.660
	*p* = 0.000	*p* = 0.000	*p* = 0.000	*p* = 0.000

Note: *N* = 9786.

**Table 2 ijerph-18-10718-t002:** The mean values, before/after usage changes, and KW test results for the usage of different types of bags between different packaging styles before and after tougher measures.

Packaging Styles	Measure Status	Inner	Reusable	Old	Charged
G	Before	0.179	0.348	0.102	0.242
	After	0.086	0.402	0.308	0.142
	Change	−51.96%	15.52%	201.96%	−41.32%
	KW test	χ^2^ = 29.303	χ^2^ = 8.247	χ^2^ = 94.032	χ^2^ = 31.837
		*p* = 0.000	*p* = 0.004	*p* = 0.000	*p* = 0.000
NG	Before	0.945	0.283	0.078	0.286
	After	0.970	0.419	0.140	0.163
	Change	2.65%	48.06%	79.49%	−43.01%
	KW test	χ^2^ = 4.638	χ^2^ = 126.24	χ^2^ = 49.418	χ^2^ = 113.01
		*p* = 0.031	*p* = 0.000	*p* = 0.000	*p* = 0.000

Note: *N* = 9786.

**Table 3 ijerph-18-10718-t003:** The logistic regression results of influential factors on people’s usage of different bags.

Bag Type	Impacting Factors	VIF	Coefficients
Inner	Gender, *generation ***, **packaging** ***, measures ****	(—, 1.010, 1.009, 1.002)	(—, 0.445, −1.652, 0.377)
Old	*Gender ***, generation ***, packaging ***, **measures** ****	(1.007, 1.042, 1.058, 1.013)	(−0.268, 0.390, 0.581, −0.770)
Reusable	** *Gender* ** ****, generation ***, packaging *, measures ****	(1.015, 1.040, 1.047, 1.011)	(−0.706, 0.659, −0.106, −0.468)
Charged	*Gender ***, **generation** ***, packaging *, measures ****	(1.008, 1.023, 1.033, 1.007)	(0.209, −0.857, 0.284, −0.702)

Notes: Italic factors and factors superscripted with asterisk(s) are verified as significantly influential on bag usage by KW test results and regression results, respectively; * *p* < 0.05, *** *p* < 0.001; gender: 1 = male, 0 = female; generation: 1 = old; 0 = middle–young; packaging style: 1 = grocery style, 0 = non-grocery style; measures: 1 = new tougher measures; 0 = old measures; bold factors are the most powerful factor in the usage of each type of bags.

## Data Availability

The data that support the findings of this study is available upon request to the corresponding author.

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
