# Peer review of "How Do Tougher Plastics Ban Policies Modify People’s Usage of Plastic Bags? A Case Study in China"

_ijerph, 2021, doi:10.3390/ijerph182010718_

Round 1
Reviewer 1 Report
This is an interesting and timely study on the behavioral effects of plastic regulations in China. It uses original data collected in Shanghai to examine changes in plastic usage. It finds that while the overall carrier bag plastic usage decreased, there are loopholes in the design and implementation of the regulations. This study adds valuable evidence on the effectiveness of government regulations at reducing plastic usage and the lessons for further improving the system. I have a few suggestions below that I hope the authors will find useful.
The authors should discuss the various incentives that shape people’s plastic bag use behavior to better interpret the meaning of the findings. For example, what are the penalties for not complying with the plastic bag ban? What are the incentives for businesses and consumers to comply or violate the plastic bag ban? Also, the Shanghai plastic ban was introduced after the national regulations—how might this sequence affect people’s behavioral change?
Regarding research design, there needs to be more information on the selection of the grocery and non-grocery stores, such as the districts they are in and the types of customers they serve. Such information can help readers infer the generalizability of the results.
Reviewer 2 Report
The paper addresses the impact of plastic banning policies on the use of plastic bags in China.
The paper is well written and the structure is adequate.
I have some comments I would like to share with the authors, hoping they will help improve the paper quality.
- The title is somewhat inaccurate; when using the expression “a comparative study” it is expected to understand immediately what the research is comparing. As such I suggest to change the title to be more clear;
- please review the reference style, since it is not according the journal guidelines;
- the paper lacks some theoretical framing; although it is a paper clearly focused on measuring the impact of public policies, some theoretical framework is needed;
- also, the paper should try to develop a deeper argument of how the behavior being measured is in fact a consequence of the policies being implemented; you are trying to explain a micro event (consumers’ individual behavior) with a macro event (the implementation of tougher plastic banning policies);
- the discussion of results needs to be improved, since it is basically limited to the presentation of research results; the discussion needs to be more critical, (1) confronting the results obtained with previous literature on the topic, and (2) presenting possible explanations of the results obtained, especially those that do not follow previous studies;
- the references’ list should be updated: from 28 references, 13 are older than 5 years.
